# Unraveling job demand-control-support patterns and job stressors as predictors: Cross-sectional latent profile and network analysis among Italian hospital workers

Igor Portoghese[1]*, Maura Galletta[1], Georg F. Bauer[2], Gabriele Finco[1], Ernesto d'Aloja[1], Marcello Campagna[1]

**1** Department of Medical Sciences and Public Health, University of Cagliari, Cagliari, Italy, **2** Division of Public and Organizational Health/Center of Salutogenesis; Department Public and Global Health at the Institute of Epidemiology, Biostatistics and Prevention, University of Zurich, Zurich, Switzerland

* igor.portoghese@unica.it

## Abstract

The Job Demand-Control-Support (JDCS) model postulates that patterns of job demand, job control, and social support lead to eight job types that are associated with well-being and health. This study employed latent profile analysis (LPA) to identify JDCS profiles among Italian hospital workers (n = 1464) and examined the predictive roles of role clarity and negative relationships at work on profile membership. Furthermore, adopting a network perspective, this study explored the interrelationships among JDCS factors within each identified profile. The LPA results revealed four profiles: isolated prisoner, moderate strain, low strain, and participatory leader. In addition, role clarity increased the likelihood of being included in the low-strain, moderate-strain, and participatory leader profiles. In contrast, negative relationships at work increased the risk of being included in the isolated prisoner profile. Finally, the results of network analysis revealed that networks differed across profiles in terms of density (interconnections between nodes) and edge strength (magnitude of relationships between nodes). Our study extends previous JDCS research by highlighting that researchers should consider empirically identified profiles rather than theoretically defined subgroups. The implications for stress theory, future research, and practice are also discussed.

## Introduction

The Job Demand-Control (JD-C) model [1] provides a well-established theoretical framework for understanding work-related stress. It posits that the level of stress experienced by workers is determined by the imbalance between job demands (i.e., the "psychological stressors involved in accomplishing the workload", p. 291; JD) and job control (meaning the "employees' freedom to use specific job skills at work

**Data availability statement:** The data analyzed in this manuscript have been posted in Open Science Framework (osf.io) and can be accessed from the following link: https://osf.io/czs29.

**Funding:** The author(s) received no specific funding for this work.

**Competing interests:** The authors have declared that no competing interests exist.

and exercise autonomy in task-related activities" p. 291; JC). Karasek and Theorell [2] extended the model by adding a third component, social support at work, defined as "overall levels of helpful social interaction available in the workplace from both co-workers and supervisors" (p. 69). Therefore, the JDCS model postulates that the work environment is determined by a combination of JD, the degree of control workers have in meeting those demands, and the support they receive from supervisors and colleagues. Moreover, both theorizations postulate a constellation of job types based on combinations of different levels (high or low) of JD, JC and job support (JS).

In his first theorization of the JD-C model, Karasek [1] hypothesized four different constellations of job types: high-strain jobs (high JD, low JC), active jobs (high JD, high JC), low-strain jobs (low JD, high JC), and passive jobs (low JD, low JC). The model was later expanded to include four additional constellations considering JC and JS combinations: participatory leader (high JC and high JS), obedient comrade (low JC and high JS), cowboy hero (high JC and low JS), and isolated prisoner (low JC and low JS) [2]. Karasek [1] suggested that employee well-being, health, and learning outcomes are linked to these job types, postulating two central hypotheses: (1) the strain hypothesis, where both high-strain and isolated-strain jobs expose workers to a greater risk of reduced psychological well-being and poor health, and (2) the learning hypothesis, which suggests that the combination of high JD and high JC (active jobs) can foster learning, motivation, and individual development at work.

Essentially, each job type represents a pattern of work environment conditions that defines different clusters of working experiences and their relationship with well-being. Despite the extensive use of the JDCS model, a significant gap in literature is related to the empirical identification of these job types and the exploration of the complex interplay among JD, JC, and JS. Previous research on JDCS model has been dominated by the variable-centered approach, examining these job types using median or percentile split methods (above or below the median\percentile) which can lead to arbitrary and potentially misleading results [3]. For example, Karasek and colleagues [4–6] defined extreme values for demand and decision latitude using tertiles. In this oversimplification, job types are considered predetermined subgroups rather than empirically derived constructs. Identifying these job types could increase our ability to tailor interventions and fully understand the implications of the JDCS model for worker well-being.

To address this limitation, this study employed latent profile analysis (LPA) to empirically identify distinct subgroups of workers based on their JD, JC and JS patterns. Also, network analysis is used to investigate the relationships among these variables within each identified profile. By considering job stress as a complex system, network analysis can help to uncover patterns of interdependence that may not be evident through traditional statistical methods. This approach may offer new insights into the mechanisms underlying work stress and well-being, facilitating the development of targeted interventions that consider the specific interplay of factors contributing to the reduction of stress in each different profile.

## Person-centered approach

Hofmans, Wille, and Schreurs [7] emphasized that occupational research has traditionally relied on variable-centered methods (e.g., ANOVA, multiple regression), which focus on linear relationships between variables across individuals. Morin, Bujacz, and Gagnè [8] argued that this approach is based on the assumption that individuals come from a homogeneous population, estimating variables as "averaged" parameters. This assumption oversimplifies data and underestimates the possibility that individuals may exhibit unique patterns of behavior, attitudes, or work experiences, potentially forming distinct latent subgroups. Thus, variable-centered approaches may fail to capture the complex relationship of variables, limiting the efficacy of organizational interventions. In contrast, a person-centered approach relaxes the assumption of homogeneity, enabling researchers to investigate unobserved heterogeneity and identify distinct configurations of variables. In this sense, variable-centered approach emphasizes the discovery of general laws and principles that apply across individuals, whereas person-centers approach emphasizes "how variables are configured to identify different subpopulations"(p. 454) [9]. This approach may help identify subgroups of workers who may benefit from targeted interventions. By identifying latent subgroups of workers with similar patterns of variables, researchers can uncover complex patterns and interactions often missed by traditional variable-centered methods. Among person-centered methods, latent class analysis (LCA) and latent profile analysis (LPA) are finite-mixture models that identify unobserved groups from observed data [10]. Individuals are probabilistically assigned to subpopulations sharing similar patterns, allowing for an empirical estimation of latent profiles\classes.

According to Wang and Hanges [11], adopting latent class procedures is crucial because they can help researchers "develop better theories and answer novel research questions" (p. 25). Despite the limited use of the person-centered approach in occupational research [12], it may offer an opportunity to expand our understanding of Karasek's model, particularly in investigating whether all the theoretically proposed job types are empirically confirmed. Over the past decade, scholars have suggested that individuals may not effectively fit into one of the eight job types described by the JDCS model. Instead, individuals may be grouped into latent profiles representing distinct subgroups of individuals who share similar JD, JC, and JS patterns [13–16]. Accordingly, these studies have moved from a variable-centered approach to a person-centered approach to identify subpopulations (latent classes/profiles) of workers with different variable patterns [17].

In line with the explorative nature of this study, we aimed to understand how the different dimensions of the JDCS model are interrelated within each profile. Network analysis [18] offers an innovative and promising approach for this purpose. Liu et al. [19] suggested that "networks have more advantages in exploring indicators' relationships" (p. 4420), offering the possibility to graphically represent the relationship among interacting profile indicators. Network analysis can be a useful approach for understanding the complex interplay of factors that contribute to job stress. Work-related stress can be conceptualized as an emergent phenomenon within complex systems, where organizational and individual factors mutually influence each other within a network of interacting components. Therefore, the overall goal of this study was to (1) apply LPA to empirically identify JDCS job types and (2) use a network approach to explore the relationships among JDCS factors within each identified profile.

## Previous studies using latent profile analyses

Despite the recent call for adopting person-centered approaches in occupational research [20], the application of this methodology within the field is considered to be "still in its infancy." Furthermore, studies adopting this approach have not consistently identified all theorized job types, providing only partial support for the JDCS model. For example, Charzyńska et al. [21] identified five profiles, whereas Portoghese et al. [16], Knight et al. [22], Urbanaviciute and Lazauskaite-Zabielske [23], Min and Hong [24], and Marzocchi et al. [25] identified four profiles. Finally, Santos et al. [26] and Galbraith et al. [27] identified three profiles. Most of these studies considered different profile indicators and different

theoretical models (JDC, JDCS, and JD-R), limiting the consistency of the results. However, two core profiles commonly identified emerge from these studies: the first, which is consistent with the high-strain job type, and is characterized by high job demands and low job control/resources; and the second, which is consistent with the low-strain job type, and is characterized by low job demands and high job control/resources.

## Predictors of latent profiles

The second objective of the current study was to explore how predictors may shape profile membership. To the best of our knowledge, research investigating predictors of JDCS profile membership remains limited. The present study investigated two potential predictors: role clarity and workplace conflicts.

According to Kahn's et al. [28] role stress theory, role stress refers to the pressure workers face when they have limited understanding of their work-related rights and obligations. That theory postulates that role stress comprises three key facets: (1) role ambiguity (unclear work expectations), (2) role conflict (mutually incongruent or incompatible work expectations), and (3) role overload (excessive work expectations). As a corollary of that theory, role clarity, conversely, is considered the antithesis of role ambiguity. It exists when "employees know the rights, duties, and responsibilities of their job, have procedural knowledge to fulfil the responsibilities, and know the consequences of [their] role" [29]. Roles within organizations are seen as a set of prescriptions that define the behaviors required of workers. Clearly defined organizational roles help employees understand the boundaries of their responsibilities, enabling them to autonomously allocate resources and effectively manage their jobs. Thus, role clarity provides a crucial organizational context for understanding objectives, prioritizing tasks, and applying skills appropriately. By defining expectations and facilitating effective action, role clarity likely shapes workers' experiences and perceptions of their workload (demands), autonomy (control), and possibly their support system [30].

Therefore, within the LPA framework, workers with high role clarity may have a higher likelihood of belonging to JDCS profiles characterized by low strain. Conversely, workers with low role clarity may have a higher likelihood of belonging to less favorable profiles, such as high-strain JDCS profiles.

The other predictor of JDCS profile we considered is social stressors, defined as "poor social interactions with direct supervisors, coworkers, and others" [31] (p. 562). Cooper and Marshall [32,33] in their stress at work model identified interpersonal relationships at work (e.g., conflicts at work) as one of the five crucial sources of occupational stress. Social stressors, such as conflicts, threats, and bullying, are commonly experienced by workers and account for 27% of all work-related stressors [34]. In their systematic review and meta-analysis, Gerhardt et al. [35] found that workplace social stressors are associated with both high burnout and low well-being. Furthermore, they suggested that social stressors at work entail large taxonomies of concepts, such as interpersonal conflict, bullying, workplace incivility, workplace aggression, or abusive supervision. All share the same common characteristic: "they represent a threat to both social self-esteem and personal self-esteem".

In line with the conservation of resource theory (COR) [36,37], social stressors, such as negative workplace relationships, may represent a threat that harms the quality of the social work environment, triggering a resource loss spiral negatively impacting on employee working attitudes, behaviors, and mental health [14].

Despite the large evidence in the variable-centered framework that social stressors play a crucial role in shaping workplace environments, studies investigating their role in a person-centered approach are lacking [14,16]. Adopting a person centered approach, Portoghese et al. [16] found that social stressors (e.g., negative relationships at work and coworker incivility) increase the likelihood of belonging to a high-strain profile. In this sense, negative relationships at work may increase perceived job demands, reduce perceived job control, and undermine positive social support. Thus, within the LPA framework, working environments characterized by high negative relationships at work, may increase the likelihood of belonging to high strain profiles. Conversely, working environments characterized by low negative relationships may increase the likelihood of belonging to low-strain profiles.

## Network analysis

The application of network theory and techniques has increased in psychometrics, particularly in the field of psychopathology, where researchers have identified network structures among symptoms [18]. Network analysis, particularly in network psychometrics, offers a complementary, variable-centered perspective. Indeed, instead of placing latent variables as the common cause of the covariation of observed variables, network models conceptualize and visualize psychological constructs (such as stress, in our study) as systems of interacting variables. Furthermore, Burger et al. [38] defined a network as a set of nodes connected by edges representing their associations. In psychological networks, nodes represent observed variables, and edges indicate the strength of associations between two variables, typically after controlling for other variables.

Network analysis has been used to study mental disorders, such as depression, anxiety, post-traumatic stress, and health-related quality of life [18]. To the best of our knowledge, the adoption of network analysis in the occupational/organizational field has been limited to a few studies investigating relationships among work engagement, work addiction, and perceived stress [39–41], among job crafting, job resources, and job demands [42,43], and between burnout and mental health [44,45]. In this context, examining the interplay of stress factors can significantly contribute to the study of work-related stress. Following Gruen et al. [46], stress can be viewed as a complex system comprising mutually influencing individual and organizational variables. Conceptualizing the interactions between these variables as a network—where variables represent nodes and interactions represent connections—provides an empirical framework to expand our understanding of how various work stress predictors are interconnected. This network approach facilitates the identification of central factors, the mapping of pathways, and the uncovering of complex patterns of conditional dependencies.

Furthermore, combining LPA and network analysis can enhance researchers' understanding of work-related stress. LPA helps identify different stress profiles by examining unique patterns of demands and resources among workers. This information can then be analyzed using network analysis to explore the specific structure and dynamics of stress-related variables within each latent profile (i.e., unobserved subgroups of workers). This integrated approach allows for the identification of specific mechanisms within each profile, leading to a more nuanced understanding of the heterogeneity of work-related stress. Additionally, this combination of methods may enable the generation of targeted hypotheses and the development of personalized interventions, potentially yielding more effective workplace strategies than using either method in isolation. For instance, recent studies [47,48] have combined LPA and network analysis for investigating network of symptoms (e.g., related to HIV or chemotherapy) among latent profiles of patients. In their study, Zhou et al. [49] investigated latent profiles of academic procrastination among nursing students and the structure of procrastination networks within each identified profile.

To advance the understanding of the JDCS model, this study addresses the lack of research on the network structure of job demands, job control, and job support within latent profiles. We adopt an exploratory approach, integrating latent profile analysis with network analysis to map the relationships between JDCS factors and their connection with role clarity and social stressors. This innovative method may provide a comprehensive framework for identifying central and bridging factors within each profile, offering new insights into the multidimensionality of the JDCS model. The resulting network map could reveal unique relationships between pairs of job factors for each identified profile, going beyond the general definition of JDCS model to investigate how specific components relate to each other within the specific system (profile). For example, network analysis can be used to compare the structure within profiles, comparing a "High Strain" profile with a "Low Strain" profile in terms of job demands, control, and support, and identifying which specific variable is influential. In turn, that may provide a more detailed understanding of the relationships in the most at-risk group...

## Study aims

First, this study aims to understand the underlying structure of the JDCS constellation of job types through the lens of latent profile analysis. Specifically, the aim of this study was to empirically identify distinct subgroups of workers (profiles) based on combinations of job demands, control, and job support. The second aim was to analyze the relationships

between these profiles and two work-related factors (role clarity and negative relationships) and then explore the network structure of variables within each identified profile.

## Materials and methods

### Study design, data collection, and participants

A cross-sectional survey study was conducted in two Italian public hospitals between June 01, 2021, and March 31, 2022. Data were collected using an online questionnaire on the LimeSurvey platform and shared via intranet and e-mail. The survey homepage provided information about the study, a general description of the questionnaire, the risks and benefits of participation (voluntary participation), and the survey's privacy policy (totally anonymous, no IP address recorded, and no personal data required). All participants provided electronic informed consent before participation by checking a box ("I understand and agree") on the homepage. A total of 1464 employees completed the survey. To ensure complete anonymity of the study, demographic data, such as gender and age, were not collected. Our study was conducted in accordance with the Declaration of Helsinki and national ethics regulations and was approved by the local ethics committee (PG/2020/11023).

### Measures

The Italian version of the HSE Management Standards Indicator Tool subscales [50,51] were used to assess JD (8 items; e.g., I have to work very fast), JC (6 items; e.g., My working time can be flexible), managerial support (MS; 5 items; e.g., I can rely on my line manager to help me out with a work problem), co-worker support (CS; 4 items; e.g., If work gets difficult, my colleagues will help me), role clarity (RC; 5 items; e.g., I am clear what is expected of me at work), and negative relationships at work (REL; 4 items; e.g.,: Relationships at work are strained). All items were rated on a 5-point Likert scale ranging from 1 (strongly disagree\never) to 5 (strongly agree\always). McDonald's ω for each subscale were: JD = 0.89, JC = 0.85, RC = 0.82, REL = 0.80, MS = 0.88, and CS = 0.88.

### Statistical analyses

**Measurement model.** The preliminary measurement models were assessed using weighted least squares mean and variance adjusted (WLSMV) in Mplus 8.1 [52]. A series of confirmatory factor analyses (CFA) and exploratory structural equation modeling (ESEM) were performed to assess the psychometric properties of the measurement model and compared the following three competing measurement models: (1) a single-factor CFA model, (2) a two-factor CFA model where JD, RC, and REL loaded in a latent variable (demands), and JC, MS, and CS loaded in a second latent variable (resources) (3) a two-factor ESEM model with a configuration similar to that of the previous model, (4) a six-factor CFA model, and (5) a six-factor ESEM model with target rotation to minimize biases in structural parameter estimates [53]. Following Xia and Yang [54], model fit has assessed using the chi-square (χ2), comparative fit index (CFI), Tucker-Lewis Index (TLI), root mean square error of approximation (RMSEA), and weighted root mean square residual (WRMR). A CFI > 0.90 and TLI > 0.95 indicate an adequate model fit, whereas values exceeding 0.95 for both indicate an excellent fit. Values smaller than 0.08 or 0.06 for the RMSEA indicate acceptable and excellent models fit, respectively, as were WRMR for values ≤ 1.00.

**Latent profile analysis (LPA).** LPA was used to extract profiles according to JD, JC, MS, and CS levels. The factor scores from the final first-order (CFA or ESEM) models were used as the profile indicators for LPA.

One to eight latent profiles were estimated using a robust maximum likelihood estimator (MLR). To prevent suboptimal local maxima, LPA employs 5000 random sets of start values, 1,000 iterations, and 500 best solutions for final-stage optimization [55].

Two general criteria has been considered when deciding how many profiles to retain: (1) consistency with the theory and conformity of the extracted profiles [8] and (2) statistical appropriateness of the extracted solution [56]. The following

goodness-of-fit indices has been considered [10,57]: Bayesian information criterion (BIC), Akaike information criterion (AIC), and constant AIC (CAIC). The bootstrapped likelihood ratio (BLRT) p-value and the Lo et al., [58] adjusted likelihood ratio (LMR-A) were used to compare the current model with the k-1 profile model. Then, we considered the entropy, where a higher value indicates greater separation between the profiles. Furthermore, the information criteria were graphically plotted as "elbow plots." Finally, we investigated the associations between the profiles and external variables. Specifically, the R3STEP command in Mplus has been used to regress latent profiles on role stress and social stressors. This command generates odds ratios (ORs) that reflect "the change in likelihood of membership in a target profile versus a comparison profile associated for each unit of increase in the predictor" [59](p. 246).

**Network analyses.** Network analyses were conducted using R version 4.3.3 [60] and visualized using the qgraph 1.9.8 package [61]. We used the identified latent profiles to define subgroups for network comparison, including role clarity and relationships at work. Because LPA does not definitively assign individuals to profiles but provides a probability distribution between profiles for each person, we assigned each individual to the profile with the highest posterior probability (>.80).

Following Fried et al.'s [62] four-step procedure, networks were estimated for each latent profile: (a) network estimation, (b) network stability, (c) network inference, and (d) network comparison, followed by Burger et al. 's reporting standards for psychological network analyses of cross-sectional data [38].

**Network Estimation**. Networks were jointly estimated using the fused graphical lasso (FGL) method. The EstimateGroupNetwork 0.3.1 package [63] was employed to obtain a layout for visualizations. To identify clusters of nodes within the four networks, we used a spin-glass algorithm implemented in the igraph 2.0.2 package [64].

**Network Stability**. 1000 bootstrap samples and nonparametric bootstrapping were used to investigate the stability of the networks using the bootnet 1.6.0 package [65]. The correlation stability coefficient was also used as a measure of network stability, which represents the maximum proportion of cases that can be dropped. A correlation stability coefficient of 0.50 or above is considered good stability, and a correlation stability coefficient of 0.25 or above is regarded as acceptable stability [65].

**Network Inference**. Node centrality, defined as the extent to which a node is more "central" than others [66], was estimated according to node strength. The bridge strength, a modified version of the node strength, was used [39]. It is defined as "a metric equal to the sum of absolute values of all edges of a given node to all other nodes that represent different psychological phenomenon" (p.11). Node predictability, defined as the proportion of variance in a node explained by all other nodes within the network ($R^2$) [67] was assessed utilizing the mgm package [68].

**Network Comparison.** The NetworkComparisonTest 2.2.2 package [69] with the seed set to 1 was used to compare pairs of networks by calculating the Spearman correlation coefficients of the edge weights for each pair. Specifically, an omnibus test was performed to examine whether all edges of the pair of networks were identical and a post hoc test (which uses the Holm-Bonferroni method to correct for multiple testing) to investigate which edge weights differed between the pair of networks. Global strength, which is the sum of all absolute edge weights for each network, was then calculated and tested for differences among networks.

After estimating a cross-sample network that considered the entire sample, the standard version of node strength, bridge strength, and node predictability were calculated. A cross-sample variability network was predicted to highlight the differences between the four networks, where each edge represents the standard deviation between the networks.

## Results

### Preliminary analysis and descriptive statistics

As a first step, factor structures of the measurement model were tested. Results suggested that the ESEM model resulted in a substantial improvement when compared to the six-factor CFA, providing a good fit to the data according

to CFI = 0.962 and TLI = 0.940, as well as an acceptable level of fit to the data according to RMSEA = 0.069 and SRMR = 0.024 (Table 1). The six *a priori* constructs were well-defined with high target factor loadings (λ = 0.33 to 0.86; M = 0.69).

## Latent profile analyses

Table 2 shows the fit indices obtained from the latent profile models containing up to 8 profiles. BIC decreased as the number of profiles increased, but the decrease became minimal, starting from the 5-profile to the 8-profile model. However, because the information criteria did not reach a minimum value, the elbow plot was inspected (see Fig 1).

Taken together, the 3-, 4-, and 5-profile solutions showed a better fit, as they were supported by the BIC and ABIC values, aLMR, and BLRT results (Table 2). Nevertheless, the 5-profile solution led to a spurious profile that comprised less than 5% of the sample, which should not be considered [55]. Considering the elbow plot, the 3- and 4-profile solutions were compared. The results were based on the BIC and SABIC values, and the aLMR and BLRT tests also supported the 4-profile solution. The LMR likelihood ratio tests showed significant results when comparing the 4-profile model with the 3-profile model (p < 0.01). In addition, this final solution was examined by considering the theoretical meaningfulness of the profiles.

Profile 1 represents 8.4% of the sample (latent profile membership probability = 0.83) and is characterized by high JD, very low JC, low CS, and very low MS (Fig 2). Following the JDCS theorization, this profile was labeled 'Isolated Prisoner'. Profile 2 represents 37.8% of the sample (latent profile membership probability = 0.83) and is characterized by slightly above average JD, slightly below average JC and CS support, and low MS. This profile was labeled 'Moderate Strain'.

**Table 1. Goodness-of-fit statistics.**

| Model | χ2 | df | CFI | TLI | RMSEA (90% CI) | SRMR |
|---|---|---|---|---|---|---|
| One-factor CFA | 17672.68 | 464 | 0.699 | 0.678 | 0.159 (0.157-0.161) | 0.116 |
| Two-factor CFA | 13957.62 | 463 | 0.764 | 0.747 | 0.141(0.139-0.143) | 0.110 |
| Two-factor ESEM | 13278.35 | 433 | 0.775 | 0.742 | 0.142 (0.140-0.144) | 0.084 |
| Six-factor CFA | 5454.30 | 449 | 0.912 | 0.903 | 0.087 (0.085-0.089) | 0.057 |
| Six-factor ESEM | 2511.97 | 319 | 0.962 | 0.940 | 0.069 (0.066-0.069) | 0.024 |

*Note*: n = 1464; χ2 = Satorra-Bentler scaled chi-square, CFA = Confirmatory Factor Analysis; ESEM = Exploratory Structural Equation Modeling; df = degrees of freedom; CFI = Comparative Fit Index; TLI = Tucker-Lewis Index; RMSEA = Root Mean Square Error of Approximation, 90% CI = 90% confidence interval for RMSEA; SRMR = Standardized Root Mean Square Residual.

**Table 2. Fit indices, entropy, and model comparisons of the estimated latent profile models.**

| Model | LL | #fp | Scaling | AIC | CAIC | BIC | SABIC | Entropy | aLMR | BLRT |
|---|---|---|---|---|---|---|---|---|---|---|
| 1-Profile | −7863.58 | 8 | 0.959 | 15743.17 | 15793.48 | 15785.48 | 15760.06 | | | |
| 2-Profiles | −7288.48 | 13 | 1.123 | 14602.96 | 14684.72 | 14671.72 | 14630.42 | 0.725 | <.001 | <.001 |
| 3-Profiles | −7181.60 | 18 | 3.197 | 14399.18 | 14512.39 | 14494.39 | 14437.21 | 0.659 | ns | <.001 |
| 4-Profiles | −7122.85 | 23 | 1.190 | 14291.70 | 14436.34 | 14413.34 | 14340.28 | 0.712 | <.01 | <.001 |
| 5-Profiles | −7102.24 | 28 | 1.391 | 14260.48 | 14436.57 | 14408.57 | 14319.62 | 0.738 | ns | <.001 |
| 6-Profiles | −7080.04 | 33 | 1.292 | 14226.07 | 14433.61 | 14400.61 | 14295.78 | 0.749 | <.05 | <.001 |
| 7-Profiles | −7061.75 | 38 | 1.239 | 14199.50 | 14438.48 | 14400.48 | 14279.77 | 0.759 | ns | <.001 |
| 8-Profiles | −7045.86 | 43 | 1.381 | 14177.72 | 14448.15 | 14405.15 | 14268.55 | 0.765 | ns | <.001 |

Note: LL = log-likelihood; #fp = number of free parameters; AIC = Akaike Information Criterion; CAIC = constant AIC; BIC = Bayesian Information Criterion; ABIC = Adjusted BIC; aLMRT = Adjusted Vuong-LoMendell-Rubin test; BLRT = bootstrap likelihood ratio test.

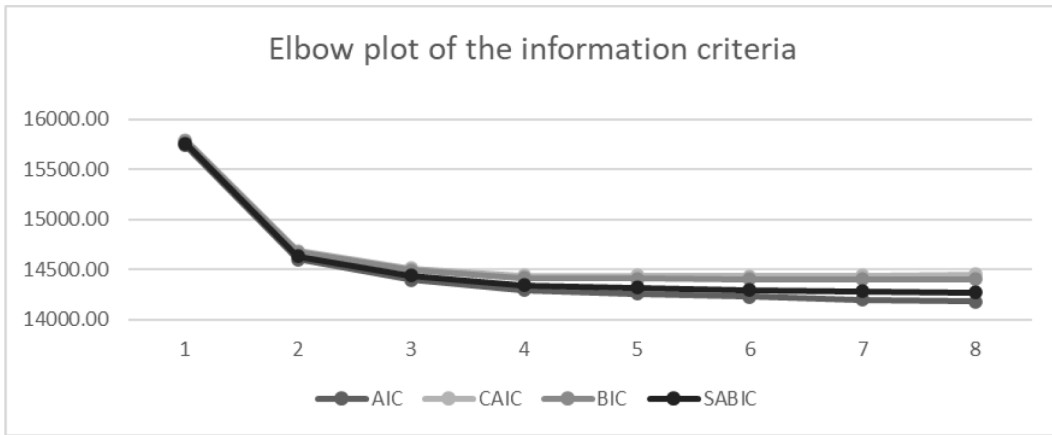

**Fig 1. Elbow plot of the information criteria.**

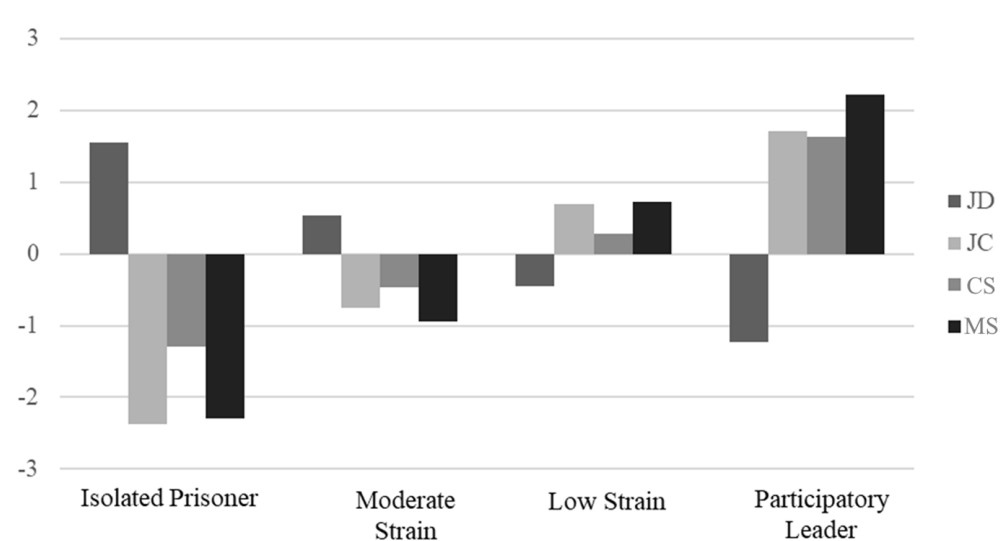

**Fig 2. Results from latent profile models (standardized values).** JD = job demands; JC = job control; MS = managerial support; CS = coworkers' support.

Profile 3 represents 44.1% of the sample (latent profile membership probability = 0.84) and is characterized by slightly below average JD and slightly above JC, CS, and MS levels. This profile was labeled "Low Strain". Finally, profile 4 represents 9.7% of the sample (latent profile membership probability = 0.82), and is characterized by low JD, high JC, high CS, and very high MS. Following the JDCS theorization, this profile was labeled 'participatory Leader'.

### Predictors of profile membership

Concerning the investigation of the antecedents of latent profiles, our results showed that role clarity and negative relationships at work increase the probability of pertaining to the profiles. We considered profile 1 (isolated prisoner) as reference for profile comparison. Results showed that role clarity increased up to 4 times the likelihood of belonging to profile 2

(OR=4.12, p<0.001), up to 11 times the likelihood of belonging to profile 3 (OR=10.76, p<0.001), and up to approximately 41 times the probability of belonging to profile 4 (OR=40.81, p<0.001).

Focusing on negative relationships at work, profiles were analyzed against a more favorable profile, which is the participatory leader category. Results showed that those working in workplaces where negative relationships are common have up to 94 times the probability of belonging to profile 1 over profile 4 (OR=94.26, p<0.001), up to 18 times the probability of belonging to profile 2 (OR=17.73, p<0.001), and up to 4 times the probability of belonging to profile 3 (OR=4.36, p<0.001).

### Stress network analyses across latent profiles and their predictors

After identifying the profiles, the network structures of each profile were compared by considering JD, JC, RC, REL, MS, and CS in our analyses. The four steps described by Epskamp et al. [65] were followed: (a) network estimation, (b) network stability, (c) network inference, and (d) network comparison.

Network Estimation. The four networks jointly estimated for the four profiles are shown in Fig 4. The network density was 0.73 (11/15 edges) for network 1 (moderate strain profile), 0.80 (12/15 edges) for network 2 (low strain profile), 0.80 (12/15 edges) for network 3 (isolated profile), and 0.87 (13/15 edges) for network 4 (participatory leader profile). The mean absolute edge weights were 0.10, 0.11, 0.13, and 0.14 for networks 1, 2, 3, and 4, respectively.

The spin-glass algorithm identified the same two clusters in the four networks. Specifically, the first cluster included job demands, role clarity, and relationships. This cluster was labeled 'Demands'. The second cluster included JC, SC, and SM. This cluster was labeled 'Resources'. The Demands cluster was linked to the Resources cluster by several consistent edges (see Fig 3): SC and SM with REL for network 1, SC and SM with REL, and JC with ROL for network 2, SC and SM with REL, and JD with JC for network 3, and SC and SM with REL for network 4.

Network Stability. The stability analyses showed that all four networks were estimated accurately, with small to moderate confidence intervals around the edge weights. All networks had correlation stability coefficients above the good stability threshold of 0.50 [65]: network1 = .75, network2 = .75, network3 = .68, and network4 = .60.

Network Inference. As shown in Fig 4, network 1's most central node was relationships at work (the unstandardized value was 0.90), and the least central node was JD (the unstandardized value was 0.43). Network 2 had the highest central node (unstandardized value of 0.91) for support from colleagues and the lowest unstandardized value (0.53) for job demands. Network 3 had the highest unstandardized value of 1.06 for relationships at work and the lowest unstandardized value of 0.43 for role clarity. Finally, for network 4, support from colleagues was the most central node (unstandardized value of 1.14), and role clarity was the least central node (unstandardized value of 0.58).

The Spearman correlation coefficients of the standard version of the node strength were 0.54 for networks 1 and 2, 0.89 for networks 1 and 3, 0.84 for networks 1 and 4, 0.31 for networks 2 and 3, 0.77 for networks 2 and 4, and 0.49 for networks 3 and 4. The predictability analysis showed that support from management was the most predictable variable (average predictability = 50.2%), and role clarity was the least predictable risk factor (average predictability = 26.4%; see Fig1). Average predictability = 22.3% in network 1, 27.6% in network 2, 53.2% in network 3, and 58.1% in network 4.

Network Comparison. In the omnibus tests of the six possible pairwise comparisons, network 1 differed significantly from network 4 (p = 0.0008), network 2 differed significantly from network 3 (p = 0.0032), and network 3 differed significantly from network 4 (p = 0.0340). A comparison of networks 1 and 4 revealed that of all 15 edges, two edges (13.3%) differed significantly: JD—ROLE (p<0.05), and SC—SM (p<0.001). The comparison of networks 2 and 3 revealed that of all 15 edges, two edges (13.3%) differed significantly: JC-REL (p<0.05), and JC-SM (p<0.001). The comparison of networks 3 and 4 revealed that of all 15 edges, three edges (20.0%) differed significantly: JD-ROLE (p<0.01), REL-SM (p<0.05), and SC-SM (p<0.01).

The global strength differed significantly ($p < 0.05$) between the four networks, and its values were 1.94, 2.17, 2.38, and 2.77 for networks 1, 2, 3, and 4, respectively. The global strength invariance test showed a significant difference between network 1 and network 4 ($p < 0.001$), network 2 and network 3 ($p < 0.001$), and network 3 and network 4 ($p < 0.05$).

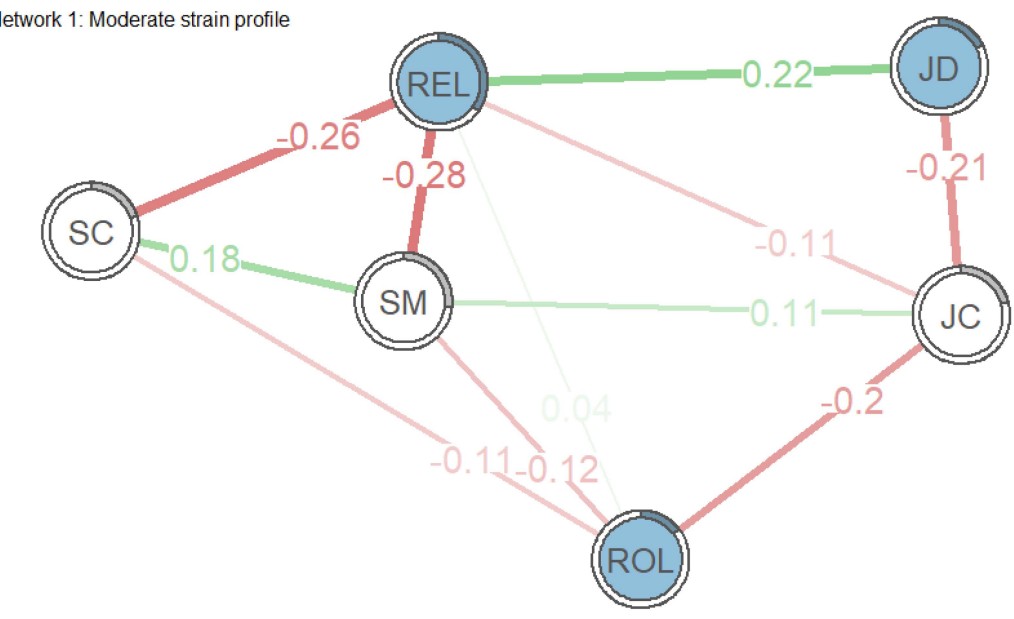

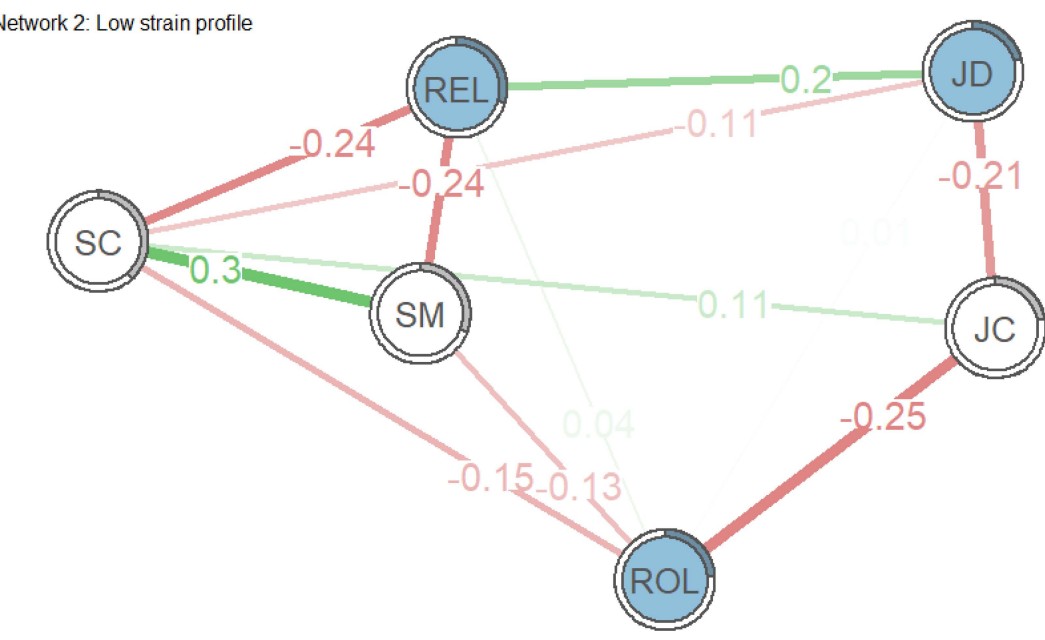

**Fig 3. Estimated network structure of latent profiles.**

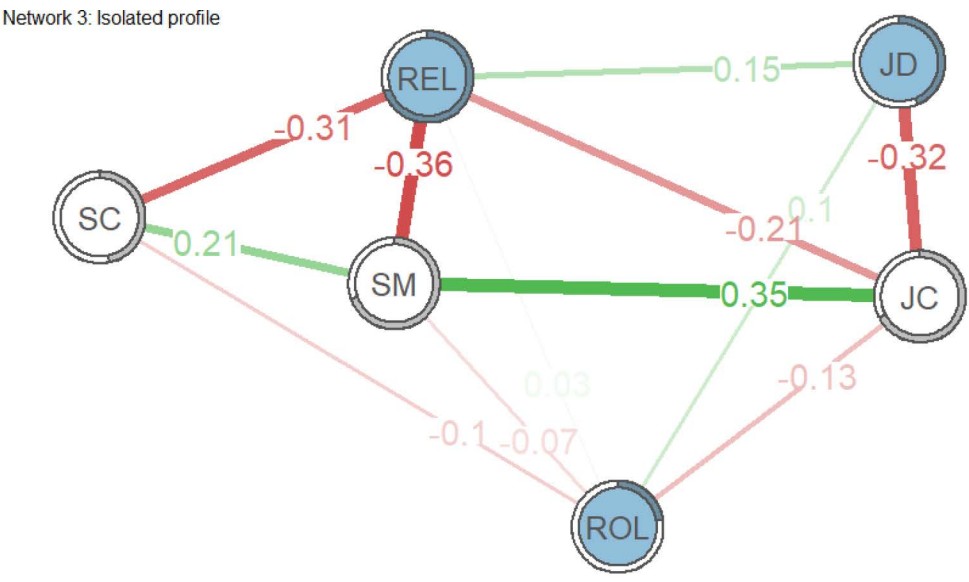

Network 3: Isolated profile

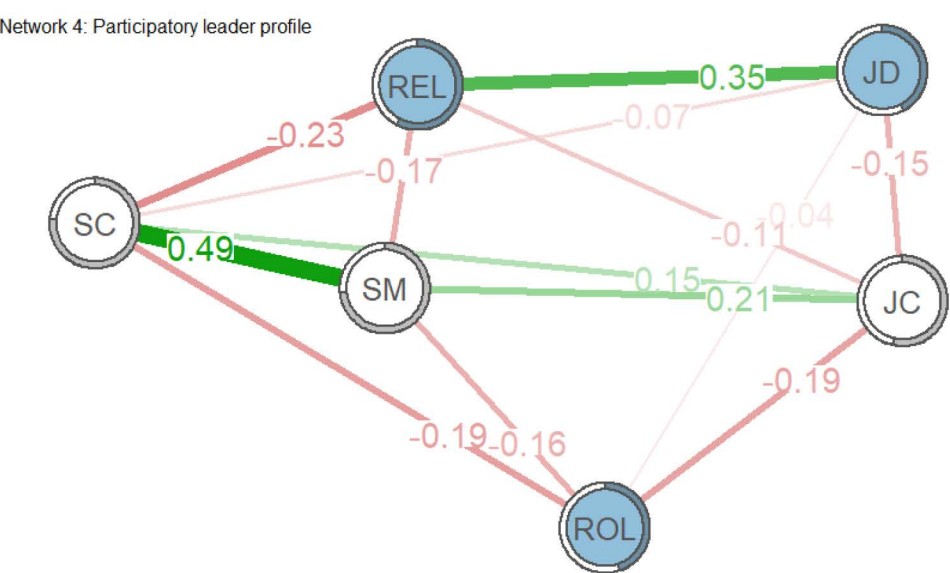

Network 4: Participatory leader profile

Note: Green and red edges indicate positive and negative partial correlations between two nodes controlled for all other nodes. The thickness of edges represents the strength of the partial correlations (the thicker the edge, the stronger the connection). Rings on nodes indicate the proportion of explained variance $R^2$; JD=job demands; JC=job control; ROL=Role; REL=negative relationships at work; SM=support from management; SC=support from coworkers.

**Fig 3.** Continued.

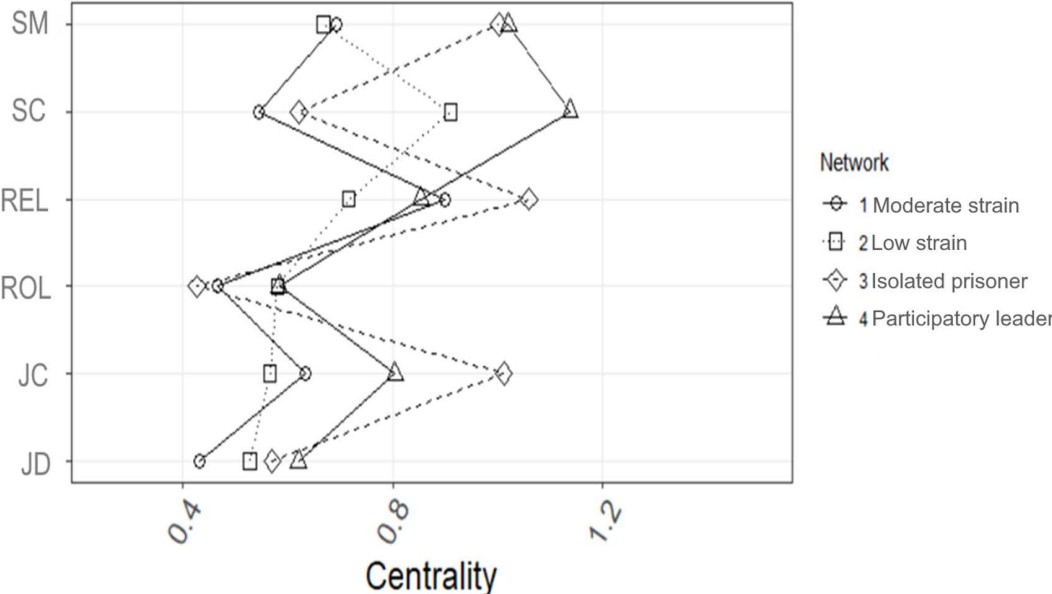

**Fig 4. Node centrality indices (standardized z scores) of the variables within each network.** JD=job demands; JC=job control; ROL=Role; REL=negative relationships at work; SC=support from coworkers; SM=support from managemen.

Fig 5 visually summarizes network properties across samples. Panel A displays the cross-sample network with the averaged edge weights, where all observations were aggregated in one sample. Panel B shows the cross-sample variability network considering the standard deviation of the edge weights among the four jointly estimated networks. Panels C and D show the unstandardized values of the node and bridge strengths, respectively, within the cross-sample network. The strongest edges connecting the variables were REL—SM (edge$_w$ = −.33), JD—JC (edge$_w$ = −.31), and SC-SM (edge$_w$ = .27). The most variable edges connecting psychosocial factors were JC-SM, SC-SM, and JC-REL, with standard deviations of 0.15, 0.14, and 0.09, respectively.

The correlation stability coefficient of the cross-sample network was 0.75, which exceeds the recommended threshold of 0.50 for the stable estimation of centrality indices [65]. The standard version of node strength revealed that management support was the most central node (unstandardized value = 0.97), and job role was the least central node (unstandardized value = 0.55).

## Discussion

Work-related stress can be considered a complex system because different components form unique configurations that follow a network structure. The JDCS model [1,2] postulates that a combination of JD, JC, and JS leads to different job type configurations. Moving from a variable-centered perspective, capitalizing on LPA can address specific research questions and expand JDCS theory regarding empirical evidence of job type configurations [20]. This study adopted an LPA perspective to identify the JDCS profiles in a sample of health care workers. Furthermore, the network structure of these profiles was investigated from a network psychometric perspective.

Our results revealed that the four profiles best represented the JDCS job types of configurations observed in the scientific literature. Only one profile matched those most frequently identified in previous studies as a low-strain profile [16,21,23,24,26,27]. Essentially, the configuration of low JD, high JC, CS, and MS might be a universal configuration, as it is common in numerous occupations [70]. Furthermore, when compared with other studies, the size of the low-strain

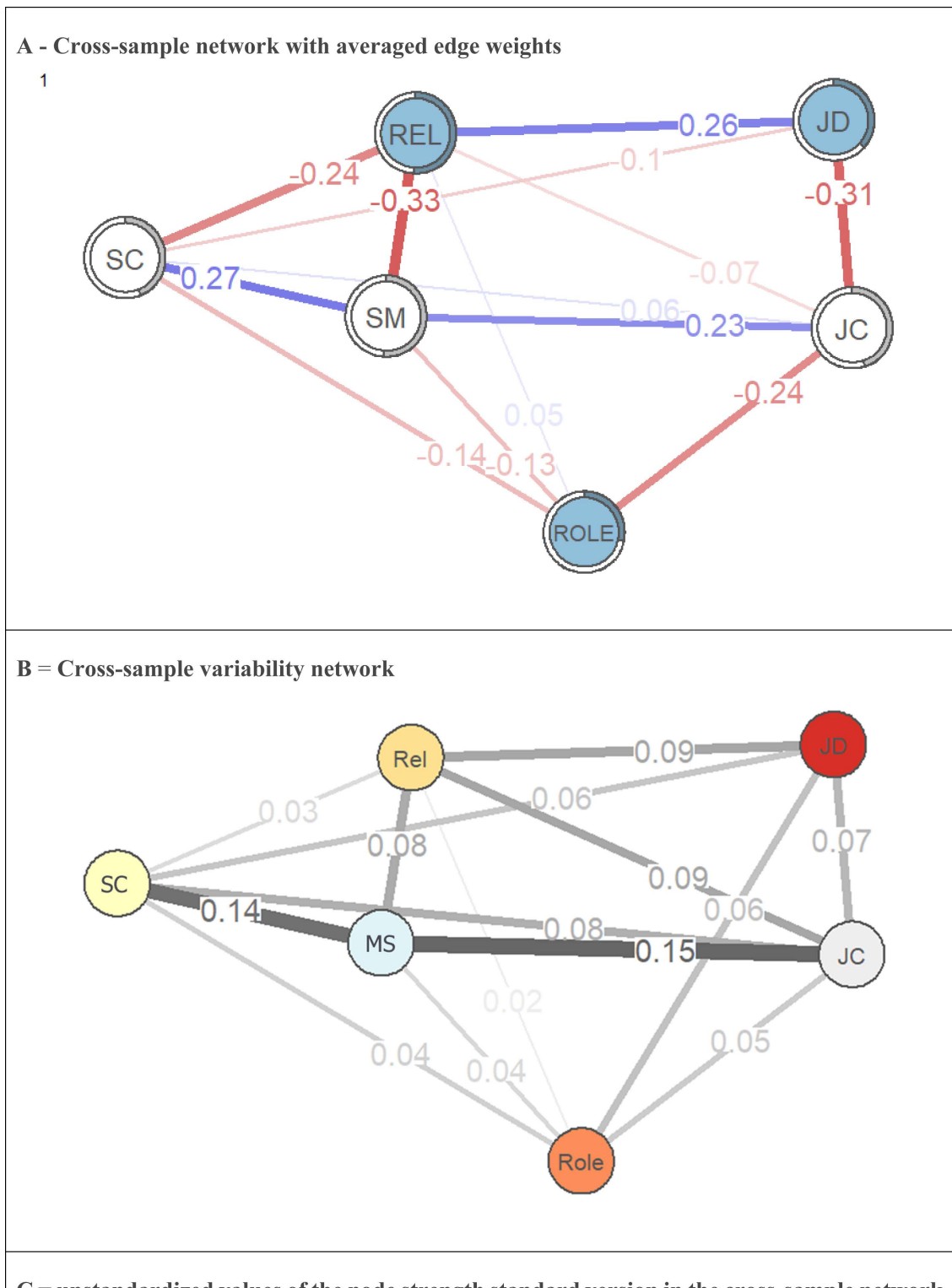

**Fig 5. Network properties across samples.** Green\red\grey edges indicate positive and negative partial correlations between two nodes controlled for all other nodes. The thickness of edges represents the strength of the partial correlations ( the thicker\darker the edge, the stronger the connection). Rings on the nodes indicate the proportion of the explained variance R2. JD=job demands; JC=job control; ROL=Role; REL=negative relationships at work; SM=support from management; SC=support from coworkers.

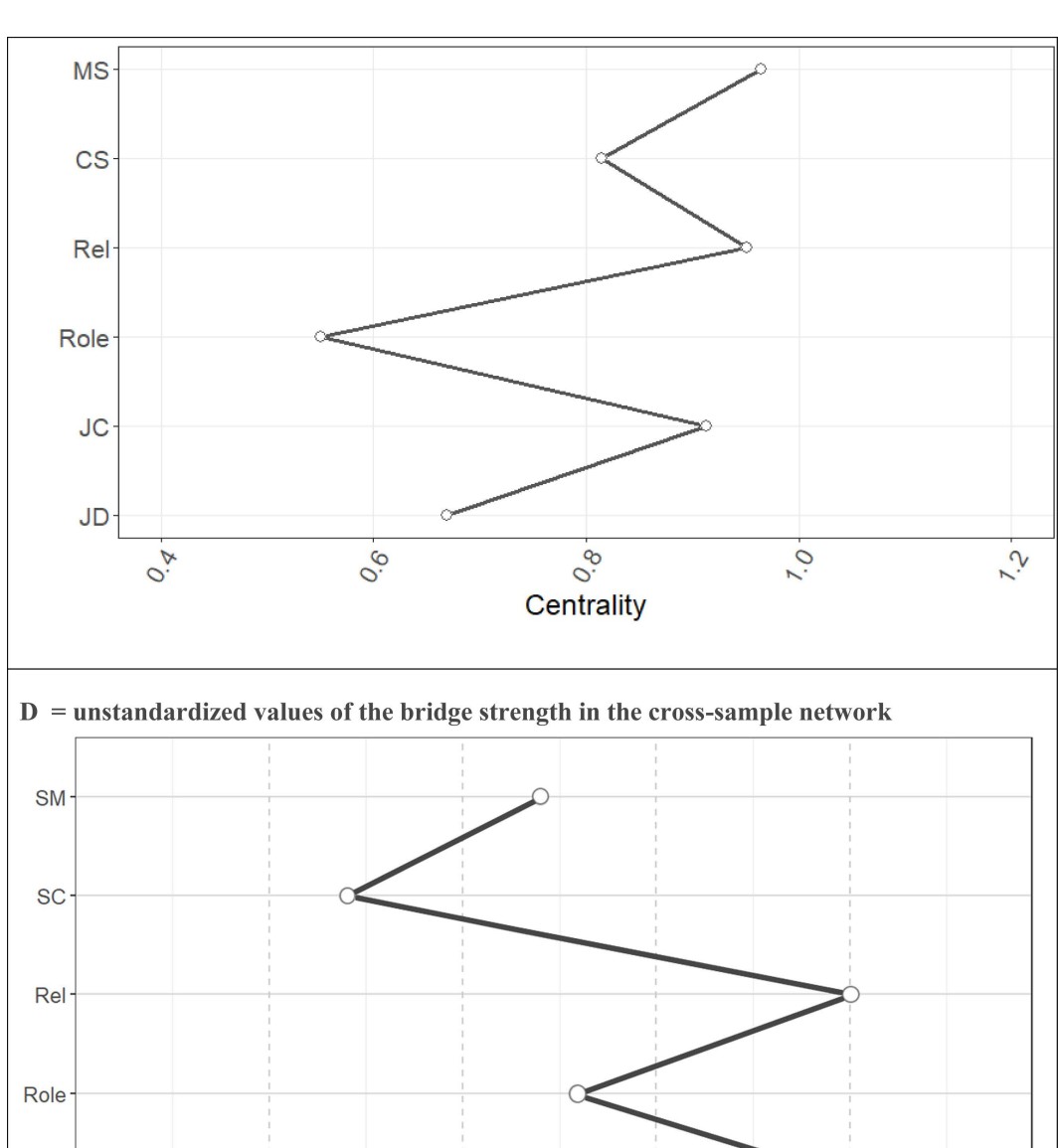

**D** = unstandardized values of the bridge strength in the cross-sample network

**Fig 5.** Continued.

profile (44.1% in our sample) is partially in line with previous person-centered approach research, reporting a profile size ranging from 10.9% to 80% in a heterogeneous sample of workers. Concerning the healthcare working population, Santos et al. [26] and Portoghese et al. [16] found profile sizes of 35.2% and 45.2%, respectively. Furthermore, Marzocchi et al. [25] identified a qualitatively different profile with low workload, low emotional dissonance, low physical and patient demands, and high control and support, labeled 'resourceful profile', representing 21.7% of the sample. Finally, Charzyńska et al. [21] reported a profile size of 18% among 516 HCWs in their analysis of JD, JC, JS, role stress, quality of working relationships, and change management.

The second profile in line with the JDCS constellation is the isolated prisoner, characterized by high JD, low JC, low CS and low MS. Our results provide empirical support for the iso-strain configuration in the JDCS model, which is in line with previous studies that investigated similar configurations [15,16,21,25,27]. This profile is an extension of the high-strain profile because it considers job support as an additional component [2]. In this sense, the high-strain and isolated prisoner configurations have the same high imbalance between JD and JC; when the JS component is considered, they represent a type of basic pattern with the first component embedded in the second. Thus, the high-strain profile can be regarded as a universal configuration, as in the case of the low-strain profile [70]. This is confirmed by most studies that considered the JS component in their LPA, such as Mäkikangas et al. [15], who found a similar profile labeled as Isolated high-strain, Portoghese et al. [16] who found the same configuration labeled as Isolated Prisoner, Charzyńska et al. [21], who found a similar profile labeled as "high stress with a good understanding of one's job role", and Galbraith et al. [27], who identified a profile designated as "High Occupational stress". Marzocchi et al. [25] identified a profile of high-strain isolated workers. In our study, a profile size of 8.3% was partially in line with previous person-centered approach research, which reported a profile size ranging from 6% to 59% in heterogeneous samples of workers. When we focus on the high-strain profile (here, not considering the support component), numerous studies [22,23,26] have shown that the prevalence of this profile ranges from 6% to 18%.

The third profile was moderate strain, characterized by slightly above-average levels of JD, slightly below-average levels of JC and co-worker support, and low managerial support. This profile is not common in the literature, but it has been identified a few times in HCW studies [16,21,71,72]. Specifically, our study is in line with Portoghese et al. [16] and Charzyńska et al. [21]. They adopted overlapping measures and indicators in the LPA, whereas Gerber et al. [71] and Jenull and Wiedermann [72] identified similar profiles among HCWs, but they considered a different pattern of variables in their model. The size of this profile (37.8%) in our study is in the middle of the range of 18%–62% in other studies.

The last profile was participatory Leader, characterized by low JD, high JC and high co-workers and managerial support. This profile is in line with the JDCS theorization and previous studies in literature. Portoghese et al. [16], Min and Hong [24], and Mazzocchi et al. [25] found similar profile configurations (low JD, high JC, and high JS), but the latter two studies labeled them as "low-strain collective" and "resourceful" respectively. Compared with other studies, the size of this profile (9.7% in our sample) was lower than the reported average profile size of 14.1% (9.6%−21.7%).

Interestingly, our study did not identify the passive and active profiles found in previous studies [22–26,73]. However, also Santos et al. [26] could not identify the passive profile of two (teachers and urban workers) of their four working population samples. Keller et al. [14] did not find active or passive profiles among their four large samples of workers, although Lee and Cho [74] did identify a similar profile labeled 'moderate resources'. When considering the healthcare working population, the passive [16,21,25,71] and active [16,21,25] profiles were not commonly identified. These profiles emerged more frequently in studies on the general working population. Future research should explore which characteristics of the working environment of HC workers can explain the low prevalence of active (high demands and high control) and passive jobs (low demands and low control) among HCWs.

Few studies have investigated predictors that may help explain JDCS profiles. Our study found connections among role clarity, negative workplace relationships, and profile membership. The results suggest general support for the role of both stressors in work-related stress literature [14,75,76], indicating that the more clear the job role, the higher the likelihood of

more favorable profiles (participatory leader and low strain profiles). Conversely, the more the workplace is characterized by negative relationships, the higher the likelihood of unfavorable profiles (isolated prisoner and moderate strain profiles). Thus, our results suggest that role clarity is associated with profiles characterized by low-level workload [77]. This is in line with Lang et al. [78], who suggested that "role clarity may be helpful for employees who experience high job demands because high role clarity results in clear expectations" (p.117). However, to the best of our knowledge, this study is the first to adopt a person-centered approach to investigate the relationship between job clarity and JDCS job types. More studies are needed to confirm this relationship.

Regarding negative workplace relationships, our results align with those of previous person-centered approach studies [14,16], suggesting that negative workplace relationships might be associated with profiles characterized by patterns of high workload. However, only Portoghese et al. (2020) considered this stressor as a profile predictor. Our study is in line with previous variable-centered studies that showed that social stressors such as tension and conflict were predictive of other job stressors [79].

After identifying the latent profiles, the network configuration of each profile was investigated. This approach offers an innovative perspective to explore how psychosocial factors are interconnected across different profiles. In particular, the community detection algorithms identified a universal two-community (cluster) structure for all four network models. The first community included JD, negative relationships at work, and role clarity, whereas the second comprised JC, peer support, and management support. In this sense, this result provides empirical evidence for the Job Demand-Resource theory [80], identifying both JD and JR communities. This study highlights the power of network analysis as a discovery tool, offering a new perspective on the underlying structure of psychosocial factors.

By examining network density, central nodes, and edge strengths, all four profiles exhibited similar network densities and overall structures, suggesting a common underlying framework for how psychosocial factors are interconnected, regardless of the specific profile. However, considering each profile, distinct central nodes (the most connected risk factors) and edge strengths (intensity of connections) were found, revealing the intricate web of relationships between job stressors within each profile.

Concerning the "moderate strain" and "isolated prisoner" profiles, both exhibited a comparable network structure with negative relationships at work constituting the central node. The "isolated prisoner" profile exhibits more intense experiences of these stressors. This suggests that strained interpersonal interactions are a key stressor, but potentially for different underlying reasons (isolation or general strain), as reflected by the different edge strengths. The profile of the 'isolated prisoner' showed stronger edge strength with support from both colleagues and management, suggesting more intense experience of these negative relationships, potentially due to isolation patterns.

Considering the more favorable profiles, "low strain" and "participatory leader," network analysis showed overlapping network structures, highlighting the importance of social support (support from colleagues) as the central node. These results confirm the importance of fostering positive work experiences in a supportive workplace. However, the "participatory leader" profile showed stronger edge strengths overall, suggesting a more robust and interconnected network of positive factors. In this sense, it is possible that all factors in the "participatory leader" profile work together more effectively to create a supportive environment. Here, network analysis reveals not only the presence of social support, but also the strength of its connections with other positive factors, providing a more complex representation of a positive work environment.

Interestingly, JD, on the other hand, appeared to be a less central node in both "low strain" and "moderate strain" profiles. This finding implies that for workers in these profiles, JD may not be as strongly interconnected with other stressors compared to the "isolated prisoner" profile. The most significant differences emerged when comparing profiles with similar overall patterns. The "moderate strain" network, while sharing some similarities with the "isolated prisoner" profile, is denser, with a stronger connection between support from colleagues and management. This finding suggests that for some workers experiencing moderate strain, greater managerial support may enhance the positive effects of colleagues'

social support. Also interestingly, this profile shows only a marginally negative relationship between job demands and role clarity. This might indicate that, for some, a clearer and less ambiguous definition of the role is not significantly related to a lower perception of high workloads, suggesting the potential need for workload management strategies alongside clarity. Finally, the 'isolated prisoner' differs from the 'participative leader' in terms of strength between SC and MS nodes, confirming that the social isolation component was crucial with respect to all other identified profiles.

## Research and practical implications

According to Hofmans et al. [81], if we want to capture the complexity of human behavior, we need develop sophisticated designs and methods. In this sense, this study advances the understanding of work-related stress within the JDCS framework, adopting two emergent methodologies. Specifically, moving beyond traditional variable-centered approaches, distinct profiles of work experience patterns were empirically identified by adopting an LPA approach. In this regard, this study challenges the traditional one-size-fits-all approach to stress assessment and intervention by offering a more nuanced perspective on the complexities of work-related stress. Furthermore, the integration of network psychometrics provides an innovative approach to unraveling the complex networks of relationships among psychosocial factors within each identified profile. In this sense, it offers deeper insight into the interplay among job demand, control, support, and other stressors.

The results of this study have significant implications for both research and practice in the field of occupational health. Researchers should move toward person-centered and network-centered approaches to further disentangle the complexity of occupational stress. This could entail exploring different working populations and employing a longitudinal approach to confirm the profiles identified in this study.

In terms of practical implications, the results of this study provide a framework for developing more effective stress-management interventions. Through the identification of distinct latent profiles and their specific network structures, organizations and managers can develop tailored interventions based on workers' profiles.

Furthermore, the network structures reveal the most central factors within each profile, guiding intervention priorities. For instance, negative workplace relationships emerged as a central node and key predictor for the moderate strain and isolated prisoner profiles. Therefore, addressing this factor is crucial for these groups, alongside prioritizing interventions that increase job control and managerial support. Specific strategies could include implementing zero-tolerance policies for bullying, harassment, and incivility, and providing training in conflict management. For the isolated prisoner profile, additional attention should be paid to interventions directly reducing social isolation and fostering workplace collaboration Concerning the participatory leader and low strain profiles, interventions should focus on maintaining high levels of job control and both forms of support. Fostering positive peer support is particularly relevant here, aligning with its centrality in these profiles' networks. Managers should also continue to actively monitor workplace relationships and promote a culture of respect.

Role clarity emerged as a protective factor, predicting membership in both the low-strain and participatory leader profiles. Therefore, organizations should actively enhance role clarity through clear job descriptions, well-defined tasks and responsibilities, and effective communication systems.

## Limitations and future directions

Our study has several limitations that should be acknowledged. First, the cross-sectional design of this study prevents us from making causal inferences about the relationships between profiles and predictors or among variables in the network analysis. Future research should adopt a longitudinal perspective to investigate the directionality of these relationships and assess the stability of profiles and networks over time. Second, although a large sample size was used, reliance on convenience samples of HCWs limits the generalizability of the findings. Future research should replicate these results

among different samples. Third, a limited set of predictors of the identified profiles was explored. Future research should investigate the role of additional factors, such as job crafting, resilience, and emotional demands, in shaping these profiles. Moreover, examining the relationships between these profiles and relevant outcomes, including well-being, work engagement, and burnout, could provide further insights into the implications of our findings for individual and organizational health. Fourth, our study focused on a specific set of psychosocial factors based on the JDCS model. Future research should consider additional stressors and resources, such as emotional demands, work-life balance, and organizational constraints, to provide a more comprehensive representation of the work environment. Fifth, the application of LPA and network analyses arise important methodological challenges. The use of modal assignments (each individual was assigned to the profile with the highest posterior probability) to create groups from probabilistic LPA classifications introduces uncertainty and potential misclassification errors, impacting network analyses. In fact, estimating networks assumes homogeneity of the conditional dependence structure within each profile. In our study, posterior classification was higher than.80 for each profile, suggesting a clear classification. However, caution is required when interpreting profile-specific networks, recognizing that observed differences could be influenced by these methodological limitations. Future studies should consider more sophisticated modeling approaches that account for classification uncertainty and allowing for heterogeneity in network structures within profiles.

Finally, it is important to remark that network analysis is exploratory. Therefore, the results should be interpreted with caution because the inclusion or exclusion of specific nodes (variables) can influence the observed network structures [82]. Future research should explore the impact of including additional variables on network configurations and investigate the robustness of our findings across different model specifications.

## Conclusions

This study identified different JDCS profiles and their related predictors. The results revealed the existence of four distinct profiles in which role clarity and social stressors predicted membership. Specifically, role clarity predicted more favorable profiles (low strain and participatory), whereas negative workplace relationships predicted stronger membership for less favorable profiles (moderate strain and isolated prisoner).

Furthermore, both LPA and network theory were used to depict multiple relationships simultaneously, providing new insights into psychosocial factors. Our research provides insights into how different resources and demands are combined for each profile. Predicting how profiles show different patterns of psychosocial factors has several important implications for the implementation of organizational job redesign interventions.

### Author contributions

**Conceptualization:** Igor Portoghese.

**Data curation:** Igor Portoghese, Maura Galletta.

**Formal analysis:** Igor Portoghese.

**Investigation:** Igor Portoghese.

**Methodology:** Igor Portoghese, Maura Galletta.

**Resources:** Igor Portoghese.

**Software:** Igor Portoghese.

**Visualization:** Maura Galletta, Georg F. Bauer, Gabriele Finco, Ernesto d'Aloja, Marcello Campagna.

**Writing – original draft:** Igor Portoghese.

**Writing – review & editing:** Igor Portoghese, Maura Galletta, Georg F. Bauer, Gabriele Finco, Ernesto d'Aloja, Marcello Campagna.

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
