## [Decision Letter · Decision Letter 0]

26 Nov 2024

Dear Dr. Portoghese,

Thank you for submitting your manuscript to PLOS ONE. After careful consideration, we feel that it has merit but does not fully meet PLOS ONE’s publication criteria as it currently stands. Therefore, we invite you to submit a revised version of the manuscript that addresses the points raised during the review process.

 2. A detailed examination of grammar, typos and other writing style weaknesses needs to be conducted and fixed.

We look forward to receiving your revised manuscript.

Kind regards,

Chunyu Zhang

Academic Editor

PLOS ONE

Journal Requirements:

3. In the online submission form, you indicated that The data underlying the results presented in the study are available from Prof. Igor Portoghese (igor.portoghese@unica.it) upon the request with reasonable causes.

Reviewers' comments:

Reviewer's Responses to Questions

**Comments to the Author**

1. Is the manuscript technically sound, and do the data support the conclusions?

Reviewer #1: Yes

Reviewer #2: Yes

2. Has the statistical analysis been performed appropriately and rigorously?

Reviewer #1: Yes

Reviewer #2: Yes

3. Have the authors made all data underlying the findings in their manuscript fully available?

Reviewer #1: Yes

Reviewer #2: Yes

4. Is the manuscript presented in an intelligible fashion and written in standard English?

Reviewer #1: No

Reviewer #2: Yes

Reviewer #1: Thank you very much for the opportunity to review interesting research. However, I have a few comments:

1. In the introduction, I am asking for a more detailed justification of the topic.

2. in the description of the material and method, add Have the studies been approved by the bioethics committee?

3. Please improve your English expression.

Reviewer #2: 1. On page 8, section 2.2, if the authors could provide one or two examples of items for different variables, it will be helpful for reading.

2. On page 11, line 252, 0.840 seems to be 0.940.

3. From line 280 on page 12 to line 289 on page 13, the sum of data 8.4%, 37.8%, 44.1%, and 9.8% is 100.1%, not 100%, which may be undesirable.

4. On page 14, line 322, "Epskamp et al., 2018" appears. This way of literature expression is inconsistent with others.

5. On page 18, line 414, it is better to change the “Participatory Leader” to “participatory leader”.

6. It is suggested that the authors provide more DOIs for the literature.

7. In the main text, the citation of literature 13 does not appear. Moreover, the order in which the literature appears should be rearranged. For instance, on page 4, after the appearance of literature 11, literature 15 follows closely, which is strange.

8. The expression "we" appears in many places in the article, which are recommended to use the neutral form.

**Do you want your identity to be public for this peer review?** For information about this choice, including consent withdrawal, please see our Privacy Policy

Reviewer #1: No

Reviewer #2: No

---

## [Author Response · Author response to Decision Letter 1]

10 Jan 2025

Dear Dr. Zhang,

Thank you for the opportunity to revise our manuscript.

We are grateful for the time and effort reviewers dedicated to reviewing our manuscript. Comments and suggestions were insightful and greatly improved the clarity and overall quality of our paper. We have carefully addressed each point, and we provide a detailed response below.

Each Reviewers’ points are reported in bold followed by our answers.

Editor's Comments:

1. A bit more depth is needed in terms of the literature review with a particular emphasis needed on the most up-to-date (years 2023-2024) management theories in the area so that you can build the agreement a bit more for why your manuscript is original by filling a gap in the research and how the paper makes a significant contribution.

(lines 70-115) - This is an interesting point that needs some clarification. At this point, no recent management theories have been developed on job stress and latent profile analysis. In this sense, we focused on the research gap on these topics. We have expanded the literature review strengthening the rationale for our study and highlights its contribution to the field by demonstrating how our person-centered approach can address gaps in understanding job stress within the context of the JDCS model.

2. Grammar and Writing Style: We have thoroughly reviewed the manuscript for grammar, typos, and writing style issues. We have corrected all identified errors and improved the overall clarity and flow of the text.

Reviewer #1:

1. Justification of the Topic: We have expanded the introduction to provide a more detailed justification of the topic, emphasizing the limitations of traditional variable-centered approaches and the potential benefits of using a person-centered approach to understand job stress profiles (lines 70-115).

2. Ethics Statement: We have added a statement in the Methods section confirming that the study was approved by the relevant bioethics committee (lines 219-221).

3. English Expression: We have carefully reviewed the manuscript for English language and expression issues and have made necessary corrections to improve clarity and accuracy.

Reviewer #2:

1. Examples of Items: We have provided examples of items for different variables (lines 223-228) to enhance reader understanding.

2. Typographical Error: We have corrected the typographical error (line 309) 0.840 to 0.940.

3. Data Sum: We have reviewed the data and corrected the sum to 100% on page 12, lines 280-289.

4. Citation Consistency: We have ensured consistency in the citation of literature throughout the manuscript, including correcting the citation on page 14, line 322.

5. Capitalization: We have corrected the capitalization of "Participatory Leader" to "participatory leader" on page 18, line 414.

6. DOIs: We have added DOIs for all cited literature where available.

7. Citation Order and Missing Citation: We have reviewed and corrected the order of citations in the text and have added the missing citation for literature 13.

8. Use of "We": We have reviewed the use of "we" throughout the manuscript and have replaced it with the neutral form where appropriate.

We believe that these revisions have significantly strengthened the manuscript and addressed all concerns raised by the reviewers.

Sincerely,

Igor Portoghese

---

## [Decision Letter · Decision Letter 1]

15 Apr 2025

Dear Dr. Portoghese,

We look forward to receiving your revised manuscript.

Kind regards,

Chunyu Zhang

Academic Editor

PLOS ONE

Journal Requirements:

Reviewers' comments:

Reviewer's Responses to Questions

**Comments to the Author**

Reviewer #3: (No Response)

Reviewer #4: All comments have been addressed

Reviewer #5: (No Response)

2. Is the manuscript technically sound, and do the data support the conclusions?

Reviewer #3: Yes

Reviewer #4: Yes

Reviewer #5: Yes

3. Has the statistical analysis been performed appropriately and rigorously?

Reviewer #3: Yes

Reviewer #4: Yes

Reviewer #5: Yes

4. Have the authors made all data underlying the findings in their manuscript fully available?

Reviewer #3: Yes

Reviewer #4: Yes

Reviewer #5: Yes

5. Is the manuscript presented in an intelligible fashion and written in standard English?

Reviewer #3: Yes

Reviewer #4: Yes

Reviewer #5: Yes

Reviewer #3: This is an interesting paper that applies latent profile analysis and network analysis to explore job demand-control-support patterns. Overall, the paper is well-written and well-structured, contributing new knowledge to the explored topic. However, I have several minor points that should be addressed before publication:

• In the Ethics Statement subsection, the Authors state: “Anonymity was assured because no IP address was registered, and no sensitive data were requested.” I assume the authors meant "personal data" rather than "sensitive data." For a study to be truly anonymous, no personal data should be collected, regardless of its sensitivity.

• Furthermore, the Authors write: "We can share only the anonymized dataset due to ethics rules." Please note that anonymous data cannot be anonymized. I assume the Authors aim to protect participant anonymity by not providing sociodemographic or work-related variables, which, when combined, might compromise anonymity in some cases. If this is the case, the authors should clarify this explicitly.

• In the Abstract, it is unclear whether the full sample (N = 1,464) was used for the network analysis.

• There is an inconsistent capitalization in the title (why are "job" and "hospital" capitalized?).

• I noticed some typos in the manuscript—the Authors are requested to correct them.

• The utility of network analysis for this study should be discussed in more detail. While the Authors apply network analysis alongside latent profile analysis, it would be helpful to elaborate on why this approach was particularly useful for understanding job demand-control-support patterns. Specifically, the manuscript should clarify what unique insights network analysis provides beyond traditional statistical methods as well as latent profile analysis.

• The Authors state: “Despite the recent call for adopting person-centered approaches in occupational research [20], the application of this methodology is still in its infancy.” However, I would not characterize person-centered approaches as being in their infancy. Please refer to Spurk et al. (2020) for a review on this topic.

• It would be beneficial to explain in more detail why role clarity and workplace relationships were chosen as predictors of latent profiles.

• When describing the number of profiles identified in previous studies, the authors write:

“For example, Charzyńska et al. [21] identified five profiles, whereas Portoghese et al. [16], Knight et al. [22], Urbanavičiūtė and Lazauskaitė-Zabielskė [23], Min and Hong [24], and Marzocchi et al. [25] identified four profiles. Finally, Santos et al. [26] and Galbraith et al. [25] identified three profiles.”

However, since different indicators (both quantitative and qualitative) were used in these studies, they are not directly comparable without reservations. This should be clearly stated in the manuscript, or the authors are encouraged to provide more details on the cited studies.

• The sentence “We considered profile 1 (isolated prisoner) to be the least favorable reference for profile comparison” is unclear. The Authors are asked to rephrase it for clarity.

• Finally, I would appreciate a more detailed discussion of the practical implications of this study.

Reference

Spurk, D., Hirschi, A., Wang, M., Valero, D., & Kauffeld, S. (2020). Latent profile analysis: A review and “how to” guide of its application within vocational behavior research. Journal of Vocational Behavior, 120, Article 103445. https://doi.org/10.1016/j.jvb.2020.103445

Reviewer #4: The authors have made the necessary and sufficient corrections in the revision of the paper. I detect no technical or formal difficulties in the new version.

Reviewer #5: The authors have effectively addressed reviewer comments. The rationale for conducting the study and the gap in the literature it addresses is well-argued in the introduction, and the paper is quite well-written. Only a few minor typographical issues remain:

Figure 2: acronyms in the legend do not match the acronyms in the note below the figure.

Line 226-228: 'has' should be replaced by 'have' (2 instances), and there is an unnecessary left parentheses before citation [51].

Line 553: "wea" should read "a"

**Do you want your identity to be public for this peer review?** For information about this choice, including consent withdrawal, please see our Privacy Policy

Reviewer #3: No

Reviewer #4: No

Reviewer #5: **Yes: ** Andrew Arena

---

## [Author Response · Author response to Decision Letter 2]

13 May 2025

Dear Dr. Zhang,

Thank you for the opportunity to revise our manuscript. We are grateful for the time and effort reviewers dedicated to reviewing our manuscript. We found reviewers’ comments and suggestions very insightful.

We have carefully addressed each point, and we provide a detailed response below.

Reviewer #3 (R3):

(R3): In the Ethics Statement subsection, the Authors state: “Anonymity was assured because no IP address was registered, and no sensitive data were requested.” I assume the authors meant "personal data" rather than "sensitive data." For a study to be truly anonymous, no personal data should be collected, regardless of its sensitivity.

 Authors reply (AR): We agree with the reviewer’s comment, and we changed it as suggested.

(R3): Furthermore, the Authors write: "We can share only the anonymized dataset due to ethics rules." Please note that anonymous data cannot be anonymized. I assume the Authors aim to protect participant anonymity by not providing sociodemographic or work-related variables, which, when combined, might compromise anonymity in some cases. If this is the case, the authors should clarify this explicitly.

 (AR): We updated the data availability statement as follows:

“The data analyzed in this manuscript have been posted in Open Science Framework (osf.io) and can be accessed from the following DOI: 10.17605/OSF.IO/NV6DK.”

(R3): In the Abstract, it is unclear whether the full sample (N = 1,464) was used for the network analysis.

 (AR):We moved the sample detail to the beginning of the sentence. All analyses were performed using the entire sample (N = 1,464).

(R3): There is an inconsistent capitalization in the title (why are "job" and "hospital" capitalized?).

 (AR): We fixed it. Thank you for carefully reading our manuscript.

(R3): I noticed some typos in the manuscript—the Authors are requested to correct them.

 (AR): We have carefully proofread the manuscript and fixed any typos.

(R3): The utility of network analysis for this study should be discussed in more detail. While the Authors apply network analysis alongside latent profile analysis, it would be helpful to elaborate on why this approach was particularly useful for understanding job demand-control-support patterns. Specifically, the manuscript should clarify what unique insights network analysis provides beyond traditional statistical methods as well as latent profile analysis.

 (AR): We agree with the reviewer’s point. The rationale behind the integration of LPA and network analysis was weak. We rewrote the paragraph by clarifying the unique value of integrating LPA and network analysis in job stress. In addition, as we have placed more emphasis on network analysis, we have added an additional network-related limitation in the discussion of limitations to the study. It is important for the reader to understand the exploratory nature of our study and the possibility that the results may be flawed by methodological limitations. Too much emphasis on adopting this methodology could create the false expectation that this is an analysis without limitations.

(R3): The Authors state: “Despite the recent call for adopting person-centered approaches in occupational research [20], the application of this methodology is still in its infancy.” However, I would not characterize person-centered approaches as being in their infancy. Please refer to Spurk et al. (2020) for a review on this topic.

 (AR): We rephrased the sentence highlighting that the use of that methodology within occupational research is still in its infancy.

(R3): It would be beneficial to explain in more detail why role clarity and workplace relationships were chosen as predictors of latent profiles.

 (AR): We agree with the reviewer’s comment. Thus, we expanded this section, adding the rationale behind this choice. However, it is important to highlight that our study was explorative in its nature and a very limited number of studies have considered predictors of JDCS profiles.

(R3): When describing the number of profiles identified in previous studies, the authors write:

“For example, Charzyńska et al. [21] identified five profiles, whereas Portoghese et al. [16], Knight et al. [22], Urbanavičiūtė and Lazauskaitė-Zabielskė [23], Min and Hong [24], and Marzocchi et al. [25] identified four profiles. Finally, Santos et al. [26] and Galbraith et al. [25] identified three profiles.”

However, since different indicators (both quantitative and qualitative) were used in these studies, they are not directly comparable without reservations. This should be clearly stated in the manuscript, or the authors are encouraged to provide more details on the cited studies.

 (AR): We agree with the reviewer’s point. We added a sentence that remarks that: “Most of these studies considered different profile indicators and different theoretical models (JDC, JDCS, and JD-R), limiting the consistency of the results.”

(R3): The sentence “We considered profile 1 (isolated prisoner) to be the least favorable reference for profile comparison” is unclear. The Authors are asked to rephrase it for clarity.

 (AR): To avoid confusing the reader, we have removed “be the least favorable” from the sentence

(R3): Finally, I would appreciate a more detailed discussion of the practical implications of this study.

 (AR): We have rewritten the practical implications of this study.

We believe that these revisions have significantly strengthened the manuscript and addressed all concerns raised by the reviewers.

Sincerely,

Igor Portoghese

---

## [Editor Report · Decision Letter 2]

15 May 2025

Unraveling job demand-control-support patterns and job stressors as predictors: Cross-sectional latent profile and network analysis among Italian hospital workers

PONE-D-24-37457R2

Dear Dr. Portoghese,

We’re pleased to inform you that your manuscript has been judged scientifically suitable for publication and will be formally accepted for publication once it meets all outstanding technical requirements.

Kind regards,

Chunyu Zhang

Academic Editor

PLOS ONE
---

## [Editor Report · Acceptance letter]

PONE-D-24-37457R2

PLOS ONE

Dear Dr. Portoghese,

I'm pleased to inform you that your manuscript has been deemed suitable for publication in PLOS ONE. Congratulations! Your manuscript is now being handed over to our production team.

Kind regards,

on behalf of

Dr. Chunyu Zhang

Academic Editor

PLOS ONE